# Long-term variations of arterial stiffness in patients with obesity and obstructive sleep apnea treated with continuous positive airway pressure

Louis-Marie Galerneau[1,2]*, Sébastien Bailly[1,2], Jean-Christian Borel[1,2], Ingrid Jullian-Desayes[1,2], Marie Joyeux-Faure[1,2], Meriem Benmerad[1,2], Marisa R. Bonsignore[3,4], Renaud Tamisier[1,2‡], Jean-Louis Pépin[1,2‡]

1 HP2 Laboratory, INSERM U1042, University Grenoble Alpes, Grenoble, France, 2 EFCR Laboratory, Thorax and Vessels, Grenoble Alpes University Hospital, Grenoble, France, 3 Division of Respiratory Medicine, PROMISE Dept, University of Palermo, Palermo, Italy, 4 IBIM CNR, Palermo, Italy

‡ Co-senior authors.
* LMgalerneau@chu-grenoble.fr

**Data Availability Statement:** All relevant data are within the manuscript and its Supporting Information files.

## Abstract

### Background

Obstructive sleep apnea (OSA) is associated with cardiovascular co-morbidities and mortality. Arterial stiffness is an independent predictor of cardiovascular risk and mortality, and is influenced by the presence of OSA and related comorbidities. There is a paucity of data regarding long-term evolution of arterial stiffness in CPAP-treated OSA patients. *We aimed to prospectively study long term PWV variations and determinants of PWV deterioration.*

### Methods

In a prospective obese OSA cohort, at time of diagnosis and after several years of follow-up we collected arterial stiffness measured by carotid-femoral pulse wave velocity (PWV), clinical and metabolic parameters, and CPAP adherence. *Univariate and multivariate analyses were performed in order to determine contributing factors.*

### Results

Seventy two OSA patients (men: 52.8%, median age: 55.8 years and median BMI of 38.5 kg/m$^2$) with a prevalence of hypertension: 58.3%, type 2 diabetes: 20.8%, hypercholesterolemia: 33.3%, current or past smoking: 59.7%, were evaluated after a median follow-up of 7.4 [5.8; 8.3] years. Over the period of follow-up, the median increase in PWV was 1.34 [0.10; 2.37] m/s. In multivariate analysis, the increase in PWV was associated with older age (10 extra years was associated with a 5.24 [1.35; 9.12] % increase in PWV) and hypertension (a significant increase in PWV of 8.24 [1.02; 15.57] %). No impact of CPAP adherence on PWV evolution was found.

**Funding:** This study was funded by an unrestricted grant from the French National Research Agency (ANR-12-TECS-0010) in the framework of the "Investissements d'avenir" program (ANR-15-IDEX-02), the "e-health and integrated care" Chair of excellence of the University Grenoble Alpes Foundation and the endowment fund "Agir pour les maladies chroniques". This study was funded in part by ORKYN Society and Périmètre Association.

**Competing interests:** LM. Galerneau, S. Bailly, I. Jullian-Desayes, M. Joyeux-Faure, M. Benmerad, MR. Bonsignore have no conflict of interest.

**Abbreviations:** AHI, apnea-hypopnea index; BP, Blood pressure; BPV, BP variability; COPD, chronic obstructive pulmonary disease; CPAP, Continuous positive airway pressure; ESS, Epworth Sleepiness Scale; FEV1, Forced Expiratory Volume in the first second of forced expiration; FVC, Forced Vital Capacity; hsCRP, high-sensitivity C-reactive protein; OSA, Obstructive sleep apnea; PWV, Pulse wave velocity; SaO2, Mean nocturnal oxygen saturation; TLC, Total lung capacity.

## Conclusion

PWV progression in CPAP-treated OSA patients is mainly related to pre-existing cardio-metabolic comorbidities and not influenced by CPAP adherence. In this high cardiovascular risk population, it is crucial to associated weight management and exercise with CPAP treatment.

## Introduction

Obstructive sleep apnea (OSA) is characterized by recurrent episodes of partial or complete obstruction of the upper airway during sleep, resulting in chronic intermittent hypoxia and sleep fragmentation. OSA is highly prevalent in obese patients with cardio-metabolic comorbidities. [1] The association between OSA and cardiovascular diseases has been clearly demonstrated, OSA being considered as an independent risk factor for cardiovascular and metabolic co-morbidities and mortality. [2–4] Continuous positive airway pressure (CPAP), the first line therapy for OSA, was reported to reduce the incidence of late cardiovascular events in patients with severe OSA in cohort observational studies. [5] However, in the largest recent randomized controlled trials, CPAP treatment did not reduce mortality or the occurrence of late cardiovascular events in intention to treat analyses. [6,7]

Arterial stiffness is an early independent predictor of cardiovascular risk and secondary occurrence of late incident cardiovascular events. [8–10] The gold standard measure of arterial stiffness is carotid-femoral pulse wave velocity (PWV).[11–13] A 1m/s increase in pulse wave velocity is associated with a 15% increase in mortality independently of other usual cardiovascular risk factors. Arterial stiffness has been suggested as having a dose-response relationship with indices of OSA severity. [8,14–16]

Arterial stiffness increases with age and blood pressure levels (BP); [17] and is linked with chronic conditions such as metabolic syndrome, [18] diabetes, [19] or chronic obstructive pulmonary disease (COPD). [20] All these conditions, which contribute to the lifelong increase in arterial stiffness, are highly prevalent in OSA patients. [21] The deterioration in arterial stiffness over time is sustained by intermediary mechanisms such as sympathetic over-activity, endothelial dysfunction, oxidative stress and systemic inflammation that are enhanced by OSA. [12,22]

There remains a debate regarding improvement in arterial stiffness under CPAP treatment. A recent meta-analysis [11] suggested an improvement but data were obtained from non-randomized studies assessing short term CPAP interventions with small sample sizes. [11,14] To date, no study has reported long term variations of arterial stiffness in CPAP-treated OSA patients. The goal of the current study was to prospectively assess the changes in PWV and their determinants in OSA patients treated by CPAP for at least four years (median duration of follow-up 7.5 years).

## Materials and methods

### Design and study population

Obese patients referred for sleep apnea to the Sleep department of Grenoble Alpes University Hospital between 2007 and 2010 were included in a prospective cohort study. These patients were re-examined after at least 4 years of CPAP treatment, with cardio-metabolic assessments including arterial stiffness. Hypertension was defined following the ESC/ESH guidelines. [23]

At inclusion, patients were aged from 20 to 75 years with a body mass index (BMI) > 30 kg/$m^2$. Patients with central apnea were excluded.

The study was conducted in accordance with good clinical practice requirements in Europe, French law, ICH E6 recommendations, and the Helsinki Declaration (1996 and 2000). The protocol was approved by an independent Ethics Committee (Comité de Protection des Personnes, Grenoble, France, IRB0006705) and registered on the ClinicalTrials.gov site (NCT02623088). All patients gave their written informed consent.

## Sleep study and sleepiness assessment

Overnight polysomnography (PSG) was used to diagnose OSA and characterize severity. [24–26] The apnea-hypopnea index (AHI) was calculated as the number of apnea and hypopnea events per hour of sleep. Daytime sleepiness was evaluated using the Epworth Sleepiness Scale (ESS). [27] Mean nocturnal oxygen saturation ($SaO_2$) and time spent under 90% of $SaO_2$ were also collected in order to characterize sleep apnea severity. Overnight sleep studies were scored according to international guidelines. [28]

## Arterial stiffness assessement

Carotid-femoral PWV, a validated measure of arterial stiffness, was assessed for each patient [12,13] using a Complior device (*Alam Medical*®, *France*). [29] Carotid-femoral PWV is the ratio on distance to transit time between two pressure waves recorded transcutaneously at carotid and femoral arterial sites. The distance travelled by the pulse wave was measured with an external tape-measure across the body surface. For the 30 min-long PWV measurements the subject was fasted and rested and in an elongated supine position. Two electrodes were placed one on the carotid artery and the other on the femoral artery until a quality signal was obtained, characterized by a clear rise of the systolic curve and a smooth diastolic curve for at least 10 seconds. At least two PWV measurements were systematically done. The mean value between the two measurements was retained if the difference between measurements was less than 0.5 m/s. When the difference was above 0.5 m/s, a third measurement was made and the median value of the three measurements was used.

## Metabolic and inflammatory biomarkers

On waking, after 10 hours fasting, a peripheral blood sample was drawn. Fasting glucose, HbA1c, serum insulin, lipids, and high-sensitivity C-reactive protein (hsCRP) levels were measured using standard procedures.

## Respiratory function

Arterial blood gas measurements and pulmonary function tests (measured using *Medisoft*® devices) were performed. Significant airway obstruction was defined as $FEV_1/FVC<70\%$, according to standard definitions. [30]

## CPAP treatment

According to French and international recommendations, [31] patients with moderate or severe OSA were treated with CPAP. [2,32] Adherence was defined as a mean CPAP use of at least 4 hours per night. [33] *CPAP adherence used for data analysis was corresponding to objective compliance measured in the 3 to 6 months preceding follow-up visit.*

### Follow-up

After 4 to 9 years of follow-up, new measurements of the same parameters as at baseline were done, except for PSG.

### Statistical analysis

Statistical analyses were performed with SAS v9.4 software (SAS Institute Inc., Cary, NC, United States). A p-value < 0.05 was considered as significant. Continuous data are presented as median and interquartile range (IQR) and categorical data as frequency and percentage. A comparison of the main quantitative variables at baseline and at follow-up was performed using a non-parametric Mann-Whitney test. A non-parametric Wilcoxon signed-rank test was used to compare the PWV before and after CPAP use. Due to the non-normality of PWV values, a log-transformation was performed and a log-linear mixed effect model with a patient random effect adjusted for the delay between the two measurements was used to analyze the evolution in arterial stiffness. A univariate analysis between PWV and potentially contributing factors was performed to select variables for the multivariate model. Variables with a p-value less or equal to 0.20 were retained and introduced into the multivariate analysis in association with predefined clinically relevant variables. Adjustment for age, sex and CPAP treatment. Due to the log transformation of the PWV, the final estimate presented in the multivariate analysis corresponded to 100*Beta (where beta was a parameter of the log-linear model and can be directly interpreted as the percent of increase or decrease in the PWV at follow-up). Due to the low number of missing values, a simple imputation method was used to impute missing data: quantitative variables were imputed using the median and qualitative variables were imputed using the most frequent value.

## Results

### Patient characteristics

As shown in the study flowchart (Fig 1), 107 obese patients were initially included in this prospective cohort. Among them, 91 were followed and treated for OSA and for 72 patients PWV was reassessed at long-term.

At inclusion, patients had a median age of 55.8 [47.4; 62.0] years, 52.8% were men, with a median (IQR) BMI of 38.5 [35.4; 43.1] kg/m$^2$. Median (IQR) AHI at diagnosis was 36.1 [23.3; 75.2] events/hour. Patients with hypertension (58.3%), had type 2 diabetes (20.8%) and were current or former smokers (59.7%). Baseline data concerning medical history, comorbidities, arterial blood gases, biological parameters, sleep studies and pulmonary function tests are shown in Table 1. The comparison between imputed and non-imputed datasets is available in S1 Table of the online supplement.

### Follow-up

The median duration of follow-up was 7.5 years. At baseline, the median value of PWV was 9.7 m/s. At the follow-up PWV assessment, the median value was 10.5 m/s corresponding to a median increase of 1.34 m/s over the follow-up period. There was a significant difference of PWV between and after CPAP use (p<0.01).

CPAP adherence of at least 4 hours/night was recorded for 72% of the patients and the median adherence to CPAP was 6.4 [5.1; 7.5] hours per night (Tables 2 & 3). The medications being used at the time of the follow-up visit are shown in S2 Table of the online supplemental material.

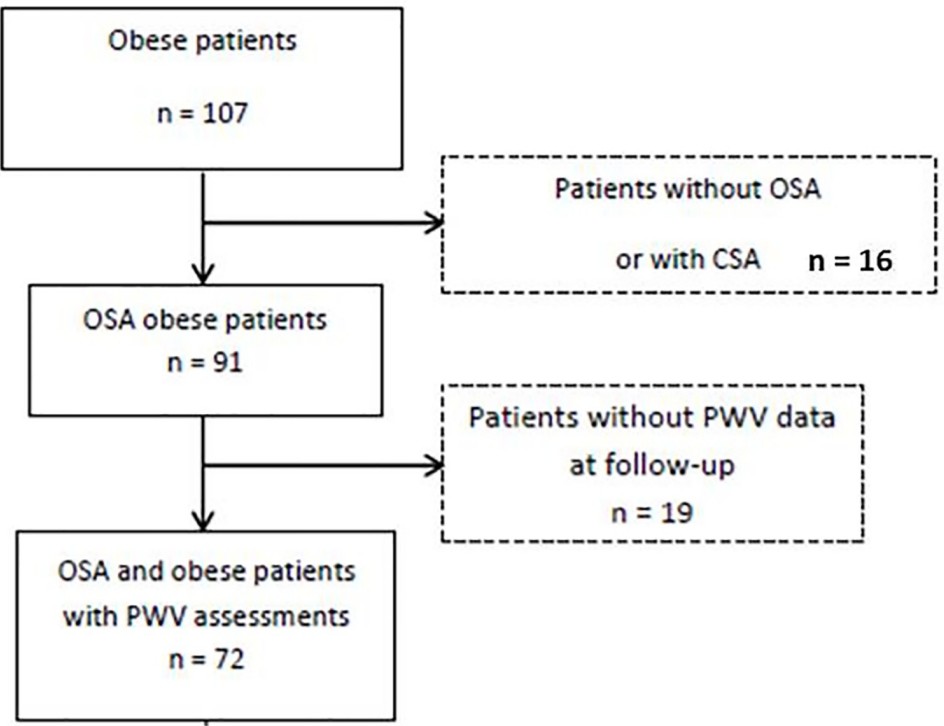

**Fig 1. Study flow chart.** CSA, central sleep apnea; OSA, obstructive sleep apnea; PWV, pulse wave velocity.

### Determinants of arterial stiffness deterioration

**Univariate analysis.**   A ten year increase in age was associated with a 7.36% increase in PWV (p<0.01). High blood pressure at baseline was associated with a 12.63% increase in PWV compared to normotensive patients (p<0.01) and having diabetes was associated with a 13.75% increase in PWV compared to patients without diabetes (p<0.01). There was no association between changes in PWV results over the years and BMI or indices of OSA severity at baseline. CPAP adherence was not linked to change in PWV. (Fig 2).

**Multivariate analysis.**   After adjustment for follow-up duration, age, gender, hypertension, diabetes, COPD, and CPAP adherence PWV was shown to increase significantly more in CPAP-treated OSA patients with hypertension (p = 0.03). A trend close to significance was apparent for type 2 diabetic patients (p = 0.08) and with airway obstruction (p = 0.11). The multivariate analysis did not demonstrate a long-term impact of CPAP adherence on PWV evolution (p = 0.54). (Fig 3).

## Discussion

To our knowledge, this is the first study assessing long-term variations (median follow-up 7.5 years) of arterial stiffness in obese OSA CPAP-treated patients. During this period, the median PWV increase under CPAP was 1.34 m/s. In multivariate analysis, PWV progression was significantly dependent of age and hypertensive status. Neither indices of OSA severity at diagnosis nor CPAP adherence contributed significantly to the long-term trajectory of arterial stiffness.

Sleep apnea is known to impact vascular age. [14,34,35] We should compare our obese OSA population (median age 55 years) to the same age group in the general population, they

**Table 1. Study population characteristics at baseline.**

| Anthropometric and biological characteristics | |
|---|---|
| Age, (years) | 55.8 [47.4–62] |
| Men | 38 (52.8) |
| BMI, (kg/m$^2$) | 38.5 [35.4–43.1] |
| Hypertension, n (%) | 42 (58.3) |
| Stroke, n (%) | 3 (4.2) |
| Diabetes mellitus, n (%) | 15 (20.8) |
| Hypercholesterolemia, n (%) | 24 (33.3) |
| Smoking, n (%) | 40 (59.7) |
| SBP, (mmHg) | 132 [122–140] |
| DBP, (mmHg) | 79.5 [70–85] |
| HbA1c, (%) | 5.8 [5.5–6.3] |
| Fasting blood Glucose, (mmol/l) | 5.7 [5.3–6.2] |
| Insulinemia, (µUl/ml) | 8.7 [6.4–13.3] |
| hsCRP, (mg/l) | 4.2 [2.1–8.9] |
| **Respiratory function** | |
| FVC, (% of predicted value) | 99 [84–106] |
| FEV$_1$, (% of predicted value) | 92 [82–103] |
| FEV$_1$/ FVC, (%) | 80.6 [75.5–84.1] |
| FEV$_1$/ FVC < 70%, n (%) | 7 (10.1) |
| TLC, (% of predicted value) | 103.5 [96.5–114] |
| PaCO2, (kPa) | 5.3 [5–5.6] |
| PaO2, (kPa) | 10.2 [9.6–11.2] |
| **Sleep disordered breathing** | |
| Epworth Sleepiness Scale | 12 [8–16] |
| AHI, (/hour) | 36.1 [23.3–75.2] |
| Mean nocturnal SpO$_2$, (%) | 92 [89–94] |
| Sleep time spent with SpO$_2$ < 90%, (% of total sleep time) | 11 [2–43] |
| **PWV (m/s)** | 9.7 [8.5–10.7] |

Categorical variables are expressed as a percentage and quantitative variables as the median (IQR). AHI, apnea hypopnea index; BMI, body mass index; DBP, diastolic blood pressure; FEV1, Forced Expiratory Volume of the first second of forced expiration; FVC, Forced Vital Capacity; HbA1c, Glycated hemoglobin; PaCO2, partial pressure of carbon dioxide; PaO2, partial pressure of oxygen; PWV, Pulse Wave Velocity; SBP, systolic blood pressure; SpO2, oxygen saturation; TLC, Total lung capacity.

**Table 2. Data at the follow-up PWV assessment.**

| | |
|---|---|
| Follow-up time, (years) | 7.5 [5.8–8.3] |
| PWV, (m/s) | 10.5 [9.6–12.7] |
| Patients adherent to CPAP, n (%) | 52 (72.2) |
| Adherence to CPAP, (hours per night) | 6.4 [5.1–7.5] |
| PWV increase during the complete follow-up period, (m/s) | 1.34 [0.10–2.37] |
| PWV increase per year, (m/s per year) | 0.19 [0.01–0.36] |

Categorical variables are expressed as percentage and quantitative variables as median (IQR). PWV, Pulse Wave Velocity.

Table 3. Comparison between baseline and follow-up values.

| Variable | Baseline | Follow-up | P |
|---|---|---|---|
| BMI (kg/m$^2$) | 38.5 [35.4; 43.1] | 38.4 [34.3; 41.9] | 0.19 |
| SBP (mmHg) | 132 [122; 140] | 132 [122; 138] | 0.44 |
| DBP (mmHg) | 79.5 [70; 85] | 74 [68; 81] | <.01 |
| hsCRP (mg/l) | 5.8 [5.5; 6.3] | 5.9 [5.7; 6.5] | 0.61 |
| Fasting Glucose (mmol/l) | 5.7 [5.3; 6.2] | 6.1 [5.4; 7] | 0.02 |
| Insulinemia (μUl/ml) | 8.7 [6.4; 13.3] | 12.2 [7.9; 16.7] | 0.08 |
| hsCRP (mg/l) | 4.2 [2.1; 8.9] | 4.1 [1.7; 5.8] | 0.22 |
| PaCO$_2$ (kPa) | 5.3 [5; 5.6] | 4.9 [4.6; 5.2] | <.01 |
| PaO$_2$ (kPa) | 10.2 [9.6; 11.2] | 11 [10.1; 11.8] | <.01 |
| Epworth Sleepiness Scale | 12 [8; 16] | 7 [4; 10] | <.01 |

BMI, body mass index; hs-CRP, high sensitivity C-reactive protein; DBP, diastolic blood pressure; PaCO2, partial pressure of carbon dioxide; PaO2, partial pressure of oxygen; SBP, systolic blood pressure.

P: p value for the non-parametric Mann-Whitney test.

probably show greater arterial stiffness at baseline, as assessed by PWV. The Arterial Stiffness Collaboration [36] reported a median (± 2 SD) PWV of 8.1 (6.3–10.0) m/s for the 50–59 year age group in the healthy population compared to 9.7 [8.5; 10.7] in our study population. A 1 m/s increase in aortic PWV corresponds to a 15% increase in all-cause mortality after adjustment for confounders. [8] This association between OSA and elevated measurements of arterial stiffness had been previously described independently of BP [15,37] or metabolic syndrome. [38] However, in a recent individual patient meta-analysis, [39] we showed that cross-sectional elevated arterial stiffness in patients with OSA is mainly driven by the conventional cardiovascular risk factors; age, BP and the presence of diabetes, while apnea severity indices had limited influence. The current data extend these results by demonstrating that long-term OSA treatment by CPAP does not check the progression in arterial stiffness.

A PWV decrease after CPAP initiation had been reported in several mostly small sample size, uncontrolled and short-term studies. [11] The largest study with a long-term follow-up showed that PWV decreased significantly over the first 6 months of treatment and then gradually increased between 6 and 24 months. [40] As in our study this late increase in PWV might be explained not only by age-related progression in arterial stiffness but also by the long-term burden of uncontrolled co-morbidities.

Hypertension is the main condition associated with PWV progression, with a reciprocal relationship between the two. [36,41–43] Severe OSA and hypertension are both associated with an increase in arterial stiffness, with cumulative effects when the two diseases coexist [14,34,44,45] In morbidly obese OSA patients CPAP has been shown to produce a small but significant reduction in blood pressure in relatively short term randomized controlled trials. [46] The SAVE study showed a non-significant systolic blood pressure difference between CPAP-treated and usual care groups of <1.0 mmHg over a mean follow-up of 3.7 years. [47] Further data on mean BP and visit-to-visit BP variability (BPV) over the first 24-months of the SAVE study have recently been reported. [48] The initial reduction in visit-to-visit BPV and mean BP was lost after 12 months and was associated with a decrease in CPAP adherence. These results are in accordance with our findings, suggesting that non-sustained reductions in mean BP and the relatively small potential effect size of CPAP are not enough to counteract the development of comorbidities and limit arterial stiffness progression. CPAP adherence was relatively high in our study population but no reduction in PWV values was observed. The

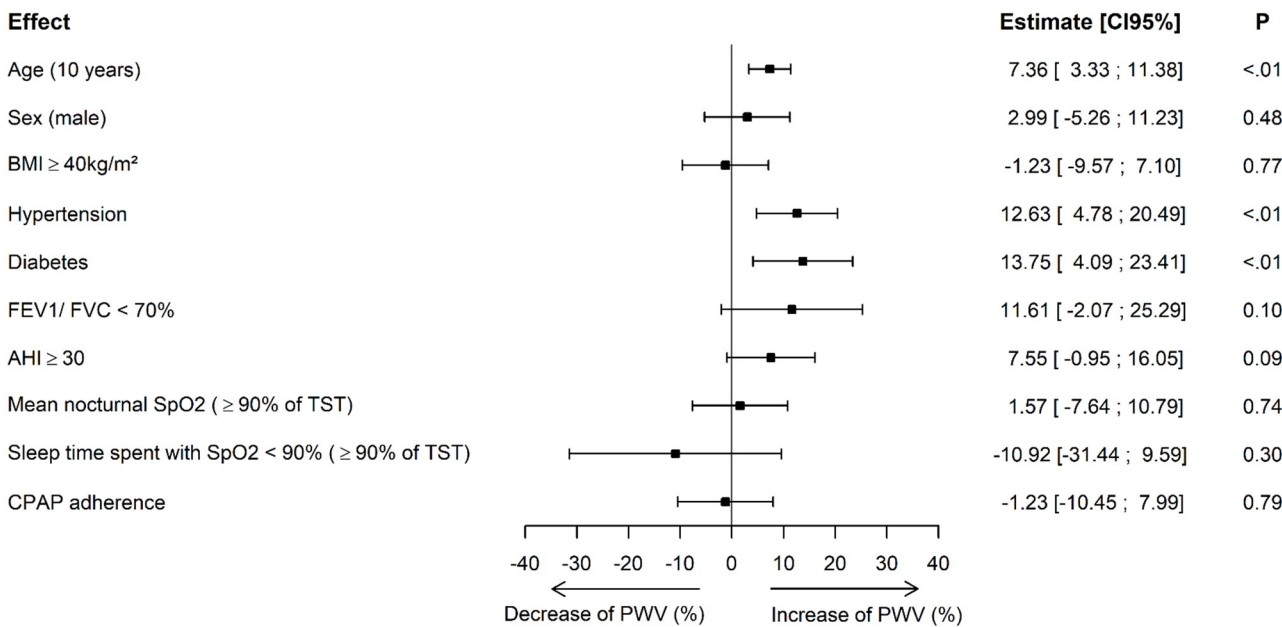

**Fig 2. Univariate analysis.** BMI, body mass index; FEV1, Forced Expiratory Volume in the first second of forced expiration; FVC, Forced Vital Capacity; AHI, apnea hypopnea index; TST, total sleep time, CPAP, continuous positive airway pressure. Interpretation: An increase of ten years in age is associated with a 7.36% increase in PWV. Having high blood pressure at baseline was associated with a 12.63% increase in PWV compared to normotensive patients. Having diabetes was associated with a 13.75% increase in PWV compared to patients without diabetes.

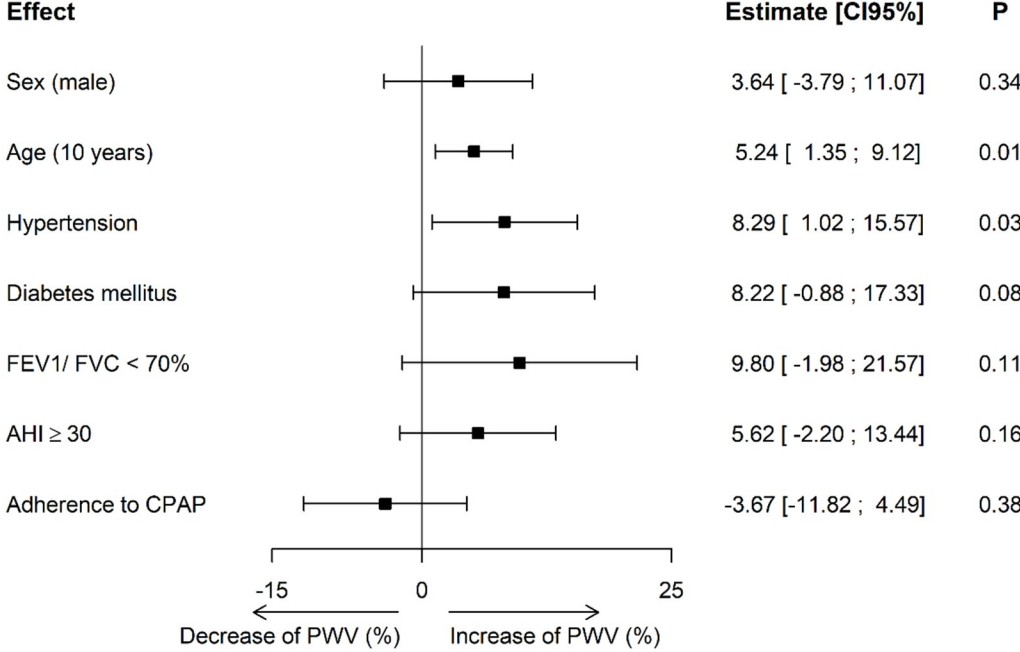

**Fig 3. Multivariate analysis.** FEV1, Forced Expiratory Volume of the first second of forced expiration; FVC, Forced Vital Capacity; AHI, apnea hypopnea index; CPAP, continuous positive airway pressure. Interpretation: A 10-year increase in age was associated with a 5.24% increase in PWV. Compared to the baseline PWV value of the multivariate model, this is associated to a significant increase of 0.35m/s of PWV for 10-year age increase. Having hypertension at baseline was associated to a significant increase in PWV of 8.24%.

follow-up long term assessment did not include a PSG as CPAP efficiency was evaluated by the index of residual events downloaded from the CPAP software's. It is unexpected that the severity of OSA changed dramatically as there was no significant change in BMI (38.5 [35.4; 43.1] versus 38.4 [34.3; 41.9] for baseline and follow-up respectively; Table 3).

Other acknowledged contributors to arterial stiffness progression are type 2 diabetes or glucose intolerance in pre-diabetic states, [19,48] and metabolic syndrome. [49] Again, OSA and metabolic syndrome synergistically act to increase PWV [38] and type 2 diabetes has a major impact toward increasing arterial stiffness in patients with metabolic syndrome. [49,50] In the present study, the association with type 2 diabetes did not reach significance in multivariate analysis, but this can certainly be explained by an insufficient sample size resulting in lack of statistical power.

The combination of COPD and OSA is called "overlap syndrome" [51] and is associated with a worse prognosis compared to that of patients with only one of the two diseases. [52–55] Our data failed to show an independent association between COPD and high arterial stiffness [20,56–58] and additive effects of COPD on the cardiovascular damage seen in patients with OSA. [59]

## Conclusion and perspectives

There is an increase in PWV over the study period. In multivariate analysis, determinants of PWV progression are old age and hypertension. Optimal management of OSA-associated comorbidities is needed for patients on CPAP treatment [60,61] in order to slow deterioration in arterial stiffness, reduce the occurrence of late cardiovascular events and to improve survival.

## Supporting information

**S1 Table. Comparison between imputed and non-imputed datasets.** AHI, apnea hypopnea index; HbA1c, Glycated hemoglobin; hs-CRP, high sensitivity C-reactive protein; DBP, diastolic blood pressure; FEV1, Forced Expiratory Volume of the first second of forced expiration; FVC, Forced Vital Capacity; SBP, systolic blood pressure; SpO2, oxygen saturation; TLC, total lung capacity.
(DOCX)

**S2 Table. Medication used by patients at the second assessment.**
(DOCX)

**S1 Data.**
(CSV)

**S2 Data.**
(CSV)

**S3 Data.**
(CSV)

## Acknowledgments

Guarantor statement: Jean-Louis Pépin takes responsibility for the content of the manuscript, including the data and analysis. Other contributions:  We thank Dr Alison Foote (Grenoble Alpes University Hospital) for critically editing the manuscript.

## Author Contributions

**Conceptualization:** Louis-Marie Galerneau, Jean-Christian Borel, Renaud Tamisier, Jean-Louis Pépin.

**Data curation:** Louis-Marie Galerneau, Jean-Christian Borel, Renaud Tamisier, Jean-Louis Pépin.

**Formal analysis:** Louis-Marie Galerneau, Sébastien Bailly, Meriem Benmerad, Renaud Tamisier, Jean-Louis Pépin.

**Funding acquisition:** Jean-Louis Pépin.

**Investigation:** Louis-Marie Galerneau, Renaud Tamisier, Jean-Louis Pépin.

**Methodology:** Louis-Marie Galerneau, Sébastien Bailly, Jean-Christian Borel, Marie Joyeux-Faure, Meriem Benmerad, Renaud Tamisier, Jean-Louis Pépin.

**Project administration:** Jean-Christian Borel, Jean-Louis Pépin.

**Software:** Sébastien Bailly.

**Supervision:** Louis-Marie Galerneau, Jean-Christian Borel, Jean-Louis Pépin.

**Validation:** Louis-Marie Galerneau, Sébastien Bailly, Jean-Christian Borel, Ingrid Jullian-Desayes, Marie Joyeux-Faure, Jean-Louis Pépin.

**Visualization:** Sébastien Bailly, Marie Joyeux-Faure, Jean-Louis Pépin.

**Writing – original draft:** Louis-Marie Galerneau, Ingrid Jullian-Desayes.

**Writing – review & editing:** Jean-Christian Borel, Marie Joyeux-Faure, Renaud Tamisier, Jean-Louis Pépin.

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
