## [Decision Letter · Decision Letter 0]

16 Mar 2020

PONE-D-20-05576

Long-term trajectories of arterial stiffness in obese CPAP-treated obstructive sleep apnea

PLOS ONE

Dear Dr Galerneau,

Thank you for submitting your manuscript to PLOS ONE. After careful consideration, we feel that it has merit but does not fully meet PLOS ONE’s publication criteria as it currently stands. Therefore, we invite you to submit a revised version of the manuscript that addresses the points raised during the review process.

Both reviewers raised several concerns that need to be addressed. Also, the reviewer listed a list of comments that require to be revised. 

We would appreciate receiving your revised manuscript by Apr 30 2020 11:59PM. To enhance the reproducibility of your results, we recommend that if applicable you deposit your laboratory protocols in protocols.io, where a protocol can be assigned its own identifier (DOI) such that it can be cited independently in the future. For instructions see: http://journals.plos.org/plosone/s/submission-guidelines#loc-laboratory-protocols

We look forward to receiving your revised manuscript.

Kind regards,

Yu Ru Kou, PhD

Academic Editor

PLOS ONE

Journal Requirements:

2. During your revisions, please clarify the wording in the title and update it in the manuscript file and online submission information. Specifically, we suggest the title to be modified as follows: "Long-term trajectories of arterial stiffness in patients with obesity and obstructive sleep apnea treated with continuous positive airway  pressure.

3. Throughout the manuscript, please avoid the use of potentially stigmatising language. For example, please replace 'obese patients' with 'patients with obesity'.

5. We note you have included a table to which you do not refer in the text of your manuscript. Please ensure that you refer to Table 3 in your text; if accepted, production will need this reference to link the reader to the Table.

Reviewers' comments:

Reviewer's Responses to Questions

**Comments to the Author**

1. Is the manuscript technically sound, and do the data support the conclusions?

Reviewer #1: Partly

Reviewer #2: Yes

2. Has the statistical analysis been performed appropriately and rigorously? 

Reviewer #1: Yes

Reviewer #2: Yes

3. Have the authors made all data underlying the findings in their manuscript fully available?

Reviewer #1: Yes

Reviewer #2: No

4. Is the manuscript presented in an intelligible fashion and written in standard English?

Reviewer #1: Yes

Reviewer #2: Yes

5. Review Comments to the Author

Reviewer #1: I’ve read with interest the manuscript entitled „Long-term trajectories of arterial stiffness in obese CPAP-treated obstructive sleep apnea”. Authors of this study compared arterial stiffness at the time of OSA diagnosis and several years later with CPAP treatment. Authors concluded arterial stiffness progresses despite CPAP treatment. I think the study is well done and the manuscript very well written. I have a few comments/suggestions:

Major:

Why was a second PSG not done during the follow up visit? Or at least the Epworth questionnaire? I think this has to be discussed as a study limitation. We do not know what happened with OSA severity of the treated subjects (it might have been worse over the years in some of the patients with not known CPAP adherence).

Have you compared those patients with known CPAP adherence of at least 4 hours with the rest? What happened with arterial stiffness in just those patients with known CPAP adherence > 4 hours?

Were there any patients with CSA? Please show more PSG parameters.

Minor:

The follow up visit was not the same for all of the patients. What was the reason?

Reviewer #2: General comment

This is an interesting article dealing with an important aspect of sleep apnea morbidity that is not study enough in the litterature

Major Comments

1. In the title, the use of « trajectories » is not appropriate. As written, the goal seems to be to draw the fluctuations of the arterial stiffness over a period. But in this study, there is a comparison of the PWV before and after at least 4 years of CPAP therapy. This term need to be changed into “variations” or “deterioration” for example

2. In the abstract

a. Precise the goal of the study: PWV variation and determinants of increased PWV

b. The methods section is not complete: write a sentence about the analysis done in the study

3. In the analysis, it is interesting and important to know if there is a statistical difference between PWV before and after CPAP therapy.

Minor Comments

Abstract

Line 50: delete “a high”

Line 51: Difficult to understand “and/or current or past smoking”

Introduction

Line 124: As said before, “long term longitudinal trajectories of arterial stiffness”

Line 126: reformulate: “assess the longitudinal changes in PWV”

Methods

Line 144: “full respiratory polysomnography”: restrict it to polysomnography, it already include the respiratory recordings

Line 145: “AHI was calculated from the number of apnea and hypopnea events per hour of sleep” : AHI was calculated “as” not “from”

Line 149 : “Overnight sleep studies were scored according to international guidelines” : precise which ones , references, they differ over years

Line 155: “the ratio of distance to transit time”: it is the ratio of distance “on” not of

Line 175: FEV1/FVC<70%, according to standard definitions : normally you should use FEV1/FVC< lower limit of the normal (LIN), this is the standard definition, according to anthropometric parameters as stipulated by GLI

Line 179-180: “Adherence was defined as a mean CPAP use of at least 4 hours per night.”: generally adherence is evaluated over a period, here was it during the 4-9 years of follow up or only during the last year before the 2nd measurement of PWV?

Line 183 : It is important to be precise about the delay after which the new PWV was measured. 4 to 9 years is too large, even more than the double the 4 years. There may be a difference in PWV according to time (4 and 9 years) in the same patients. Here precise if it was only one measurement of PWV or more after that 4 years of CPAP therapy. This part need to be more precise!!!

Line 193: “univariable analysis”: it is univariate analysis same for multivariable, It is multivariate

Line 195 : Multivariate analysis was compute via which type of regression model?

Line 196: “Age, sex and CPAP treatment were included in the final multivariable model independently of the results of the univariable analysis”: this is called ajustement for age, sex and CPAP treatment

Line 201: “Due to the low number of missing values, a simple imputation method was used to impute missing data: quantitative variables were imputed using the median and qualitative variables were imputed using the most frequent value”: why was there missing values of PWV? This was a prospective study and the primary outcome was PWV, how missing values can be explained only on 107 patients?

Line 203-204: “Statistical analyses were performed with SAS v9.4 software (SAS Institute Inc., Cary, NC, United States)”: this is at the beginning of the section statistical analysis

Line 205: A p-value < 0.05 not “A p-value of < 0.05”

Results

Figure 1:

- Put the n of patients without OSA or with CSA

- As asked before, I don’t understand how 19 patients did not have data on PWV at follow up : Was it a retrospective study?

- Why are you dividing patients at the end into 2 groups according to the CPAP adherence? It is not necessary here to tell us how patients were included in the study. The outcome is changes in PWV not the adherence to CPAP therapy

Line 212: write median (IQR)

Line 214: “Patients commonly exhibited hypertension (58.3% of the population)” rephrase: delete commonly exhibited and of the population

Line 215: remove frequently; in the methods there is no mention about how a patients was characterized as current or former smoker!

Table 1:

- It is not possible to put n (%) in column + Median (IQR) and the same n (%) in lines and at the end “Categorical variables are expressed as a percentage and quantitative variables as the median (25th, 75th percentiles)”. This is repetition!!! . Normally n (%) on the head of the column and median (25th, 75th percentiles) on line for only qualitative variables. And if it is like that, it is not necessary to add as footnotes “Categorical variables are expressed as a percentage and quantitative variables as the median (25th, 75th percentiles)”

- In the bracket of IQR, hyphen not semi column

- Add comma after the name of the variable, eg: Age, (years)- BMI, (Kg/m²)….

- FEV1/ FVC < 70% (%) ; it is n (%)

- It is Fasting “blood”Glucose

- Footnotes: FEV1 is forced expiratory volume during the first second; add SBP, PaCO2, PaO2, SpO2

Line 221:

It is “Determinants of arterial stiffness deterioration” not Determinants deterioration in arterial stiffness

The title of this section don’t fit the results presented. In that part there is a description of Arterial stiffness in the studied population. Please, give another title.

It is after that there the title “Determinants of arterial stiffness deterioration” which will include 2 sections: univariate and multivariate analysis

Line 224: the expression “over the whole period” is not appropriate

Table 2

- Commas after the name of the variable

- Footnotes: remove “a” after as; TLC is not on this table; the expression of median should be uniform over the manuscript, chose between IQR and (25th-75th percentile) even if they express the same thing

Line 230 : remove figure on the title of the section

Figure 2: it is 95% CI not the CI95% and precise the definitions of abbreviations used there somewhere

Line 232: what is the definition of hypertension in the work? is it elevated blood pressure during the investigation or a notion of Hypertension treated or not before? Important to precise it in methods. In that way we will really interpret the results.

Line 234-235: “There was a trend toward a significant association between increase in PWV and COPD (P=0.10)” not well said. The p value is superior from your cut off but this potential associated factor can be included in the final model of multivariate analysis.

Line 240: really good the adjustment

Line 243-244: “The multivariable analysis did not demonstrate a long-term impact of CPAP adherence on PWV evolution”: I am not convinced. CPAP adherence here is evaluated when? Is it the mean or median of CPAP adherence measured each year?

Line 248: To the best of our knowledge instead of to our knowledge; review the expression long term trajectories

Line 251 : dependent of not on

Line 252: “with trends suggesting the influence of type 2 diabetes and COPD comorbidities” : no, not to considered as a major result. The p value is not significant

Line 253: “Neither indices of OSA severity at diagnosis or CPAP adherence” , better to say “Neither indices of OSA severity at diagnosis nor CPAP adherence”

Line 255: “Sleep apnea is known as impacting vascular “ better to say “Sleep apnea is known to impact vascular …

Line 255-261: “our obese OSA population (median age 55 years) showed greater arterial stiffness at baseline, as assessed by PWV, compared to the same age group in the general population”: this assertion have not been verified in your results, you have not compare those groups and had a statistical significant difference. Please rephrase all that

Line 269 : Had been not has been

Line 299-301: no, data shown in this study don’t confirm independent association between COPD and PWV because the P value in multivariate analysis is not significant

The conclusion and perspective section is only talking about perspective. Add a first part that summary your main pertinent results.

Figure 3: same remarks as figure 2

6. PLOS authors have the option to publish the peer review history of their article (what does this mean?). If published, this will include your full peer review and any attached files.

Reviewer #1: No

Reviewer #2: No

---

## [Author Response · Author response to Decision Letter 0]

6 May 2020

Grenoble, March 26Th

Manuscript number PONE-D-20-05576

Dear Pr Yu Ru Kou,

Please find enclosed the revised version of the manuscript entitled « Long-term trajectories of arterial stiffness in obese CPAP-treated obstructive sleep apnea» by L-M. Galerneau et al., for consideration to be published in Plos One.

Thank you for your e-mail regarding the above manuscript. We are grateful to the reviewers for their time and effort in evaluating our manuscript, and appreciate their constructive comments and criticisms. The following is a point-by-point response to the reviewers’ comments. We provide a revised version with changes marked in red and a clean version.

Yours sincerely,

Dr Louis-Marie Galerneau Pr Jean-Louis Pépin

Answers to reviewer 

We thank reviewers for the constructive comments and criticisms. 

Comments to the Author

Reviewer #1

I’ve read with interest the manuscript entitled „Long-term trajectories of arterial stiffness in obese CPAP-treated obstructive sleep apnea”. Authors of this study compared arterial stiffness at the time of OSA diagnosis and several years later with CPAP treatment. Authors concluded arterial stiffness progresses despite CPAP treatment. I think the study is well done and the manuscript very well written. I have a few comments/suggestions:

We thank the reviewer for her/his positive appreciation of our work.

Major:

Why was a second PSG not done during the follow up visit? Or at least the Epworth questionnaire? I think this has to be discussed as a study limitation. We do not know what happened with OSA severity of the treated subjects (it might have been worse over the years in some of the patients with not known CPAP adherence).

The follow-up long term assessment did not include a PSG as CPAP efficiency was evaluated by the index of residual events downloaded from the CPAP software’s. It is unexpected that the severity of OSA changed dramatically as there was no significant change in BMI (38.5 [35.4 ; 43.1] versus 38.4 [34.3 ; 41.9] for baseline and follow-up respectively; Table 3). 

This paragraph has been added in the discussion in the study limitation section.

Or at least the Epworth questionnaire? 

We thank the reviewer for this comment. ESS data are available at follow-up and confirm CPAP efficacy. The following line has been added in table 3:

ESS at follow-up: 7 [4; 10] 

Have you compared those patients with known CPAP adherence of at least 4 hours with the rest? What happened with arterial stiffness in just those patients with known CPAP adherence > 4 hours?

This issue was already addressed in the submitted version of the paper. The variable CPAP adherence was included in the multivariate analysis (See figure 3) and had no impact on PWV trajectories.

Were there any patients with CSA? Please show more PSG parameters.

Patients with central apnea were excluded. The following sentence has been included in the methods section:

« Patients with central apnea were excluded »

Minor:

The follow up visit was not the same for all of the patients. What was the reason?

The assessments during the follow-up visit were the same for all the patients. 

Reviewer #2

General comment

This is an interesting article dealing with an important aspect of sleep apnea morbidity that is not study enough in the literature

We thank the reviewer for her/his positive appreciation of our work.

Major Comments

1. In the title, the use of « trajectories » is not appropriate. As written, the goal seems to be to draw the fluctuations of the arterial stiffness over a period. But in this study, there is a comparison of the PWV before and after at least 4 years of CPAP therapy. This term needs to be changed into “variations” or “deterioration” for example

According to this comment, the title has been replaced by “Long-term variations of arterial stiffness in obese CPAP-treated obstructive sleep apnea”

2. In the abstract

a. Precise the goal of the study: PWV variation and determinants of increased PWV

We have added in the abstract the following sentence: “We aimed to prospectively study long term PWV variations and determinants of PWV deterioration”.

b. The methods section is not complete: write a sentence about the analysis done in the study

To take into account the suggestions of the two reviewers the statistical methods paragraph has been partly rewritten (see below) and we have addressed all the comments:

Statistical Analysis

Statistical analyses were performed with SAS v9.4 software (SAS Institute Inc., Cary, NC, United States). A p-value of < 0.05 was considered as significant. Continuous data are presented as median and interquartile range (IQR) and categorical data as frequency and percentage. A comparison of the main quantitative variables at baseline and at follow-up was performed using a non-parametric Mann-Whitney test. A non-parametric Wilcoxon signed-rank test was used to compare the PWV before and after CPAP use. Due to the non-normality of PWV values, a log-transformation was performed and a log-linear mixed effect model with a patient random effect adjusted for the delay between the two measurements was used to analyze the evolution in arterial stiffness. A univariable univariate analysis between PWV and potentially contributing factors was performed to select variables for the multivariable multivariate model. Variables with a p-value less or equal to 0.20 were retained and introduced into the multivariable multivariate analysis in association with predefined clinically relevant variables. Ajustement for age, sex and CPAP treatment. Due to the log transformation of the PWV, the final estimate presented in the multivariable multivariate analysis corresponded to 100*Beta (where beta was a parameter of the log-linear model and can be directly interpreted as the percent of increase or decrease in the PWV at follow-up). Due to the low number of missing values, a simple imputation method was used to impute missing data: quantitative variables were imputed using the median and qualitative variables were imputed using the most frequent value. 

3. In the analysis, it is interesting and important to know if there is a statistical difference between PWV before and after CPAP therapy.

Yes, the difference was significant and the following sentence has been added in the results section:

 There was a significant difference of PWV between and after CPAP use (p<0.01).

Minor Comments

Abstract

Line 50: delete “a high” => done

Line 51: Difficult to understand “and/or current or past smoking” => changed

Introduction

Line 124: As said before, “long term longitudinal trajectories of arterial stiffness” => trajectories replaced by variations

Line 126: reformulate: “assess the longitudinal changes in PWV” => longitudinal has been deleted

Methods

Line 144: “full respiratory polysomnography”: restrict it to polysomnography, it already include the respiratory recordings => done

Line 145: “AHI was calculated from the number of apnea and hypopnea events per hour of sleep” : AHI was calculated “as” not “from” => changed

Line 149 : “Overnight sleep studies were scored according to international guidelines” : precise which ones , references, they differ over years => done

Line 155: “the ratio of distance to transit time”: it is the ratio of distance “on” not of => changed

Line 175: FEV1/FVC<70%, according to standard definitions : normally you should use FEV1/FVC< lower limit of the normal (LIN), this is the standard definition, according to anthropometric parameters as stipulated by GLI

Basically, we agree but FEV1/FVC<70% was one pre-specified inclusion criteria and we cannot change at this stage.

Line 179-180: “Adherence was defined as a mean CPAP use of at least 4 hours per night.”: generally adherence is evaluated over a period, here was it during the 4-9 years of follow up or only during the last year before the 2nd measurement of PWV?

We thank the reviewer for asking for clarification on this. CPAP adherence used for data analysis was corresponding to objective compliance measured in the 3 to 6 months preceding follow-up visit. This sentence has been added in the methods section.

Line 183: It is important to be precise about the delay after which the new PWV was measured. 4 to 9 years is too large, even more than the double the 4 years. There may be a difference in PWV according to time (4 and 9 years) in the same patients. Here precise if it was only one measurement of PWV or more after that 4 years of CPAP therapy. This part need to be more precise!!!

We have taken into account the delay between the two PWV measurements and results were already presented as a mean PWV increase per year (see table 2).

Line 193: “univariable analysis”: it is univariate analysis same for multivariable, It is multivariate => changed

Line 195: Multivariate analysis was compute via which type of regression model?

It was a multivariable log-linear mixed effect model with adjustment on age, sex and CPAP treatment. This has been included in the new statistical analysis section (see above, reviewer 1).

Line 196: “Age, sex and CPAP treatment were included in the final multivariable model independently of the results of the univariable analysis”: this is called adjustment for age, sex and CPAP treatment => changed

Line 201: “Due to the low number of missing values, a simple imputation method was used to impute missing data: quantitative variables were imputed using the median and qualitative variables were imputed using the most frequent value”: why was there missing values of PWV? This was a prospective study and the primary outcome was PWV, how missing values can be explained only on 107 patients?

There are technical issues for reproducible measurement of PWV in morbidly obese patients. Some data were then not available at follow-up. 

Line 203-204: “Statistical analyses were performed with SAS v9.4 software (SAS Institute Inc., Cary, NC, United States)”: this is at the beginning of the section statistical analysis => changed

Line 205: A p-value < 0.05 not “A p-value of < 0.05” => done

Results

Figure 1:

- Put the n of patients without OSA or with CSA

- As asked before, I don’t understand how 19 patients did not have data on PWV at follow up : Was it a retrospective study?

This is a prospective study registered in clinical trials. See response above regarding missing data. 

- Why are you dividing patients at the end into 2 groups according to the CPAP adherence? It is not necessary here to tell us how patients were included in the study. The outcome is changes in PWV not the adherence to CPAP therapy

This has been suppressed in the study flow chart.

Line 212: write median (IQR) => done

Line 214: “Patients commonly exhibited hypertension (58.3% of the population)” rephrase: delete commonly exhibited and of the population => done

Table 1:

- It is not possible to put n (%) in column + Median (IQR) and the same n (%) in lines and at the end “Categorical variables are expressed as a percentage and quantitative variables as the median (25th, 75th percentiles)”. This is repetition!!! . Normally n (%) on the head of the column and median (25th, 75th percentiles) on line for only qualitative variables. And if it is like that, it is not necessary to add as footnotes “Categorical variables are expressed as a percentage and quantitative variables as the median (25th, 75th percentiles)” => Only footnotes have been maintained

- In the bracket of IQR, hyphen not semi column => done

- Add comma after the name of the variable, eg: Age, (years)- BMI, (Kg/m²)….=> done

- FEV1/ FVC < 70% (%) ; it is n (%) => corrected

- It is Fasting “blood”Glucose => added

- Footnotes: FEV1 is forced expiratory volume during the first second; add SBP, PaCO2, PaO2, SpO2 => added

Line 221:

It is “Determinants of arterial stiffness deterioration” not Determinants deterioration in arterial stiffness => changed

The title of this section don’t fit the results presented. In that part there is a description of Arterial stiffness in the studied population. Please, give another title. => the title has been changed

It is after that there the title “Determinants of arterial stiffness deterioration” which will include 2 sections: univariate and multivariate analysis => this title has been replaced after

Line 224: the expression “over the whole period” is not appropriate => changed

Table 2

- Commas after the name of the variable => corrected

- Footnotes: remove “a” after as; TLC is not on this table; the expression of median should be uniform over the manuscript, chose between IQR and (25th-75th percentile) even if they express the same thing => changed

Line 230 : remove figure on the title of the section => removed

Figure 2: it is 95% CI not the CI95% and precise the definitions of abbreviations used there somewhere => changed

Line 232: what is the definition of hypertension in the work? is it elevated blood pressure during the investigation or a notion of Hypertension treated or not before? Important to precise it in methods. In that way we will really interpret the results.

Hypertension was defined as follow:

“Hypertension was defined following the ESC/ESH guidelines ». We have included this sentence in the methods section with the following reference:

 Williams LB, Mancia EG, Spiering YHW, Agabiti Rosei EE, Azizi EM, Burnier EM, et al. 2018 ESC/ESH Guidelines for the management of arterial hypertension: The Task Force for the management of arterial hypertension of the European Society of Cardiology and the European Society of Hypertension: The Task Force for the management of arterial hypertension of the European Society of Cardiology and the European Society of Hypertension. Journal of Hypertension. 2018;36(10):1953-2041.

This has been included in the methods section.

Line 234-235: “There was a trend toward a significant association between increase in PWV and COPD (P=0.10)” not well said. The p value is superior from your cut off but this potential associated factor can be included in the final model of multivariate analysis. => this sentence has been deleted

Line 240: really good the adjustment=> thanks

Line 243-244: “The multivariable analysis did not demonstrate a long-term impact of CPAP adherence on PWV evolution”: I am not convinced. CPAP adherence here is evaluated when? Is it the mean or median of CPAP adherence measured each year?

See responses above to the same comments. Thanks.

Line 248: To the best of our knowledge instead of to our knowledge; review the expression long term trajectories => ok

Line 251: dependent of not on => ok

Line 252: “with trends suggesting the influence of type 2 diabetes and COPD comorbidities” : no, not to considered as a major result. The p value is not significant => deleted

Line 253: “Neither indices of OSA severity at diagnosis or CPAP adherence” , better to say “Neither indices of OSA severity at diagnosis nor CPAP adherence” => modified

Line 255: “Sleep apnea is known as impacting vascular “ better to say “Sleep apnea is known to impact vascular … => ok

Line 255-261: “our obese OSA population (median age 55 years) showed greater arterial stiffness at baseline, as assessed by PWV, compared to the same age group in the general population”: this assertion have not been verified in your results, you have not compare those groups and had a statistical significant difference. Please rephrase all that => ok

Line 269 : Had been not has been => ok

Line 299-301: no, data shown in this study don’t confirm independent association between COPD and PWV because the P value in multivariate analysis is not significant => modified

The conclusion and perspective section is only talking about perspective. Add a first part that summary your main pertinent results. => modified

Figure 3: same remarks as figure 2 => modified

---

## [Decision Letter · Decision Letter 1]

22 May 2020

PONE-D-20-05576R1

Long-term variations of arterial stiffness in patients with obesity and obstructive sleep apnea treated with continuous positive airway pressure

PLOS ONE

Dear Dr. Galerneau,

Thank you for submitting your manuscript to PLOS ONE. After careful consideration, we feel that it has merit but does not fully meet PLOS ONE’s publication criteria as it currently stands. Therefore, we invite you to submit a revised version of the manuscript that addresses the points raised during the review process.

One reviewer still had some suggestions. Particularly, he/she did not find one sentence in the revised manuscript that was indicated in your response..

We look forward to receiving your revised manuscript.

Kind regards,

Yu Ru Kou, PhD

Academic Editor

PLOS ONE

Reviewers' comments:

Reviewer's Responses to Questions

**Comments to the Author**

1. If the authors have adequately addressed your comments raised in a previous round of review and you feel that this manuscript is now acceptable for publication, you may indicate that here to bypass the “Comments to the Author” section, enter your conflict of interest statement in the “Confidential to Editor” section, and submit your "Accept" recommendation.

Reviewer #1: All comments have been addressed

Reviewer #2: (No Response)

2. Is the manuscript technically sound, and do the data support the conclusions?

Reviewer #1: Yes

Reviewer #2: Yes

3. Has the statistical analysis been performed appropriately and rigorously? 

Reviewer #1: Yes

Reviewer #2: Yes

4. Have the authors made all data underlying the findings in their manuscript fully available?

Reviewer #1: Yes

Reviewer #2: Yes

5. Is the manuscript presented in an intelligible fashion and written in standard English?

Reviewer #1: Yes

Reviewer #2: Yes

6. Review Comments to the Author

Reviewer #1: I would like to thank the authors for revising the manuscript. I think it substantially improved. I have no further comments/questions.

Reviewer #2: Authors have addressed the major part of our recommendations.

But some few corrections need to be done:

Line 153: the reference of AASM need to be in the reference list

Line 178: if “FEV1/FVC<70% was one pre-specified inclusion criteria” you should write it in the section participant as an inclusion criteria. Also, if you write according to standard definitions, we need to have a reference.

Line 180-183: “We thank the reviewer for asking for clarification on this. CPAP adherence used for data analysis was corresponding to objective compliance measured in the 3 to 6 months preceding follow-up visit. This sentence has been added in the methods section.” I am not seeing that sentence in the section on CPAP treatment

Line 243: delete “figure 3” in the title of the section “multivariate analysis” and put it in the text below

Line 311: please, in the conclusion it is not appropriate to write details as “1.34 [0.10; 2.37] m/s” … etc. You can write: “there is an increase in PWV over the study period. Determinants of PWV progression are old age and hypertension”

7. PLOS authors have the option to publish the peer review history of their article (what does this mean?). If published, this will include your full peer review and any attached files.

Reviewer #1: No

Reviewer #2: Yes: Virginie Poka-Mayap

---

## [Author Response · Author response to Decision Letter 1]

8 Jul 2020

All the answers to the questions and the corrections of the reviewers are present in the file "Response to Reviewers".

Line 153: the reference of AASM need to be in the reference list :

 The reference has been added in the text

Line 178: if “FEV1/FVC<70% was one pre-specified inclusion criteria” you should write it in the section participant as an inclusion criteria. Also, if you write according to standard definitions, we need to have a reference. : 

FEV1 / FVC <70% was not an inclusion criteria of our study. We use FEV1 / FVC <70% to define the presence of airflow limitation on pulmonary function test. A reference to FEV1 / FVC <70% defining the airflow limitation has been added in the text.

Line 180-183: “We thank the reviewer for asking for clarification on this. CPAP adherence used for data analysis was corresponding to objective compliance measured in the 3 to 6 months preceding follow-up visit. This sentence has been added in the methods section.” I am not seeing that sentence in the section on CPAP treatment 

: Indeed, this sentence did not appear in the method section. The sentence "CPAP adherence used for data analysis was corresponding to objective compliance measured in the 3 to 6 months preceding follow-up visit" was added in the CPAP treatment subsection of the method section.

Line 243: delete “figure 3” in the title of the section “multivariate analysis” and put it in the text below :

Done

Line 311: please, in the conclusion it is not appropriate to write details as “1.34 [0.10; 2.37] m/s” … etc. You can write: “there is an increase in PWV over the study period. Determinants of PWV progression are old age and hypertension” : 

Thank you for your comment. The change has been made in the text.

---

## [Decision Letter · Decision Letter 2]

13 Jul 2020

Long-term variations of arterial stiffness in patients with obesity and obstructive sleep apnea treated with continuous positive airway pressure

PONE-D-20-05576R2

Dear Dr. Galerneau,

We’re pleased to inform you that your manuscript has been judged scientifically suitable for publication and will be formally accepted for publication once it meets all outstanding technical requirements.

Kind regards,

Yu Ru Kou, PhD

Academic Editor

PLOS ONE

Additional Editor Comments (optional):

Reviewers' comments:

Reviewer's Responses to Questions

**Comments to the Author**

1. If the authors have adequately addressed your comments raised in a previous round of review and you feel that this manuscript is now acceptable for publication, you may indicate that here to bypass the “Comments to the Author” section, enter your conflict of interest statement in the “Confidential to Editor” section, and submit your "Accept" recommendation.

Reviewer #2: All comments have been addressed

2. Is the manuscript technically sound, and do the data support the conclusions?

Reviewer #2: Yes

3. Has the statistical analysis been performed appropriately and rigorously? 

Reviewer #2: Yes

4. Have the authors made all data underlying the findings in their manuscript fully available?

Reviewer #2: Yes

5. Is the manuscript presented in an intelligible fashion and written in standard English?

Reviewer #2: Yes

6. Review Comments to the Author

Reviewer #2: (No Response)

7. PLOS authors have the option to publish the peer review history of their article (what does this mean?). If published, this will include your full peer review and any attached files.

Reviewer #2: **Yes: **Virginie Poka-Mayap

---

## [Editor Report · Acceptance letter]

20 Jul 2020

PONE-D-20-05576R2 

Long-term variations of arterial stiffness in patients with obesity and obstructive sleep apnea treated with continuous positive airway pressure 

Dear Dr. Galerneau:

I'm pleased to inform you that your manuscript has been deemed suitable for publication in PLOS ONE. Congratulations! Your manuscript is now with our production department. 

Kind regards, 

on behalf of

Dr. Yu Ru Kou 

Academic Editor

PLOS ONE